# HARMONIC CONSTRAINED REINFORCEMENT LEARNING

## ABSTRACT

Constrained reinforcement learning (CRL) aims to train agents that maximize rewards while satisfying safety constraints, an essential requirement for real-world application. Despite extensive progress in using various constrained optimization techniques, striking a stable balance between reward maximization and constraint satisfaction remains a challenge. Reward-driven updates often violate constraints, while overly safety-driven updates degrade performance. To address this conflict, we propose harmonic constrained reinforcement learning (HCRL), a framework that resolves reward-safety trade-offs at the gradient level in an optimal manner. At each iteration, HCRL formulates a trust-region minimax optimization problem to compute a harmonic gradient (HG) for the policy update. This gradient has minimal conflict with both the reward and safety objective gradients, thereby enabling more stable and balanced policy learning. In practice, we can equivalently convert this challenging constrained minimax problem for solving HG as an unconstrained single-variable optimization problem, maintaining high time-efficiency. Empirical results on three planar constrained optimization problems and ten Safety Gymnasium tasks demonstrate that HCRL consistently outperforms existing CRL baselines in terms of stability and the ability to find feasible and optimal policies.

## 1 INTRODUCTION

Reinforcement learning (RL) has achieved significant success in several areas including Chinese Go (Schrittwieser et al., 2020), video games (Vinyals et al., 2019), and robotics (Ju et al., 2022). It enables agents to acquire optimal policies through a trial-and-error mechanism designed to maximize expected cumulative rewards (Li, 2023). However, in many real-world control tasks, the optimal policy not only needs to maximize rewards but also requires strictly satisfy safety constraints (Gu et al., 2024). For example, an autonomous vehicle has to pay attention to both reaching the destination and avoiding collision simultaneously.

The cruciality of safety in real-world applications has driven the rapid development of constrained reinforcement learning (CRL), also known as safe reinforcement learning (Safe RL) (García & Fernández, 2015). A major research line is to employ constraint optimization methods to equip standard RL with constraint-handling capabilities. Alex Ray et al. incorporate the Lagrangian multiplier method into proximal policy optimization (PPO), resulting in PPO-Lagrangian, which enables both constraint satisfaction and return maximization (Ray et al., 2019). Another line of research leverages feasibility theory to jointly learn a safety value function, also called a feasibility function. This line uses both the standard reward value function and the feasibility function to design a region-wise policy improvement objective, aiming to achieve the highest reward while maximizing the feasible region. There are various choices of feasibility functions, including control barrier function (Ma et al., 2022; Ames et al., 2019), Lyapunov functions (Chow et al., 2018; Richards et al., 2018), safety index (Ma et al., 2022) and Hamilton-Jacobi reachability function (Yu et al., 2022). Regardless of the specific constraint optimization techniques used to enable RL to handle constraints, the policy will inevitably face the challenge of balancing reward and safety objectives at each training iteration. Only by optimizing both two objectives to the greatest extent simultaneously can we obtain the converged feasible and optimal policy.

As shown in Figure 1, to provide an intuitive illustration of this balance challenge, we classify the policy set into three categories based on the two dimensions (reward and safety): (1) *Worthless* (blue

area): these policies are both unsafe and low-valued, typically representing undertrained policies in the early stages of training. (2) *Risky* (red area): these policies are high-valued but unsafe, usually reflecting insufficient consideration of constraints due to the algorithm overly prioritizing reward improvement. (3) *Conservative* (green area): these policies are safe but low-valued, indicating they are overly restricted, as the agent is too cautious to pursue higher rewards. The goal of CRL is to search for the feasible and optimal policy, which maximizes rewards while satisfying safety constraints. This ideal policy often lies at the intersection of these categories, highlighting the significant challenge of striking a balance between rewards and safety.

This paper proposes harmonic constrained reinforcement learning (HCRL), which balances reward maximization and safety satisfaction at the gradient level in an optimal manner. It is achieved by first calculating two separate policy gradients, one for reward and one for safety, and then computing their harmonic gradient with minimal conflicts for policy updates through solving an optimization problem at each RL iteration.

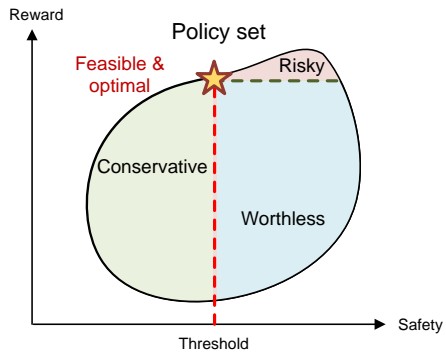

Our contributions are summarized as: (1) We are the first to introduce gradient harmonization with minimal conflict into CRL. At each RL iteration, HCRL solves a harmonic gradient (HG) by minimizing the negative impact of conflict between reward and safety objectives, thereby better approaching the desired feasible and optimal policy. (2) We employ HG to augment Lagrangian-base CRL algorithms off-policy DSAC-

Figure 1: An intuitive classification of policies based on both reward and safety.

Lag and on-policy PPO-Lag methods, yields DSAC-HLag and PPO-HLag methods. (3) Comprehensive experiments on planar optimization and safety gymnasium demonstrate that the proposed HCRL methods perform better or comparable to classical CRL baselines. It exhibits improved training stability while more effectively balancing reward maximization and safety constraint satisfaction.

## 2 RELATED WORK

**Constrained RL** has gained growing attention due to the safety requirements in the practical applications of RL. Mainstream constrained RL leverages constraint optimization techniques to equip agents with the ability to ensure safety (Gu et al., 2024). These techniques can be divided into two categories, penalty and Lagrangian methods. Thanks to the ease of implementation, penalty methods (Liu et al., 2020b; Guan et al., 2022), such as interior-point and exterior-point, were widely adopted to convert constraints as punishment terms. However, adding a punishment term into the policy objective with a fixed penalty ratio will lead to an inevitable drift of optimal solution (Smith et al., 1997).Therefore, recent research has majorly turned to the Lagrange multiplier method, which formulates a minimax optimization on the Lagrangian function involving Lagrange multipliers and policy parameters as learnable variables (Li, 2023). Representative methods include SAC-Lagrangian, PPO-Lagrangian, etc (Ha et al., 2021; Ray et al., 2019). Regardless of the specific constraint optimization techniques used to enable RL to handle constraints, the policy inevitably will face the challenge of balancing reward and safety objectives.

**Mitigating conflict between reward and safety** is a widely recognized problem that needs to be addressed for the Lagrangian-based CRL algorithms (Ray et al., 2019; Zhang et al., 2025). As a pioneering work, Stooke et al. incorporated feedback control theory and proposed a proportion-integration-differentiation (PID) Lagrangian method (Stooke et al., 2020). It can use the integral, magnitude, and derivative of the constraint violation signal to adjust the multiplier, thereby reducing oscillations significantly and achieving more balanced convergence (Sohrabi et al., 2024). SPIL (Peng et al., 2021) draws inspiration from PID control again and introduce an integral separation technique, i.e., separate the integrator out when the integral value is large to avoid a potentially integral overshooting and stable learning. The above mentioned works focus on multiplier adjustment to reduce the negative impact of conflicts. Our work orthogonally considers to eliminate the conflict between the original reward term and the safety term at the gradient level in an optimal manner, aiming at achieving better convergence and finding the balanced feasible and optimal solution.

## 3 PRELIMINARY

### 3.1 PROBLEM STATEMENT

We formulate the CRL problem as an MDP with deterministic dynamics, defined by the tuple $\langle \mathcal{S}, \mathcal{A}, P, r, c, \gamma \rangle$, where: (1) $\mathcal{S}$ and $\mathcal{A}$ are state and action spaces; (2) $P : \mathcal{S} \times \mathcal{A} \times \mathcal{S} \rightarrow \{0, 1\}$ is the dynamics; (3) $r : \mathcal{S} \times \mathcal{A} \rightarrow \mathbb{R}$ is the reward signal; (4) $c : \mathcal{S} \rightarrow \{0, 1\}$ is the cost signal, where $c = 1$ if any constraint is violated and $0$ otherwise; (5) $\gamma \in (0, 1)$ is the discount factor (Li, 2023; Yu et al., 2022). The policy $\pi : \mathcal{S} \rightarrow \mathcal{A}$ is deterministic that chooses action $a_t$ at state $s_t$ at each time $t$. The initial state distribution is $d_0(s)$. The objective of standard RL is to find a policy maximizing the expected return $J_r(\pi) = \mathbb{E}_{s_0 \sim d_0(s), \pi} \sum_t [\gamma^t r(s_t, a_t)]$. A value function $V_r^\pi(s) \triangleq \mathbb{E}_\pi \sum_t [\gamma^t r(s_t, a_t) | s_0 = s]$ represents the potential return in the future at state $s$. One can notice that $J_r(\pi) = \mathbb{E}_{s \sim d_0(s)}[V_r^\pi(s)]$. In CRL, the expected cost return $J_c(\pi)$ and cost value function $V_c^\pi(s)$ are defined and learned similarly by replacing the reward signal $r$ by the cost signal $c$ (Altman, 2021). The goal of CRL becomes to maximize $J_r$ while minimizing $J_c$ towards zero. Here we use $g_r$ and $g_c$ to represent $\nabla_\theta J_r$ and $\nabla_\theta J_c$, respectively. It is worthy to note that those policies with both $g_r$ and $g_c$ reach zero are competent candidates of the feasible and optimal policy (Xu et al., 2021).

### 3.2 LAGRANGE MULTIPLIER METHOD

The Lagrange multiplier method is a well-known constrained optimization technique. It introduces Lagrange multiplier to combine the objective function and constraint into a single scalar function called the Lagrangian, enabling simultaneous optimization of both the original decision variables and the newly introduced multiplier (Ha et al., 2021). Consider a general constrained optimization problem where $\theta$ denotes the decision variables, $J_r(\theta)$ is the objective function to be maximized, and $J_c(\theta) \leq 0$ is an inequality constrain. The Lagrangian $\mathcal{L}$ is defined as $\mathcal{L}(\theta, \lambda) = J_r(\theta) + \lambda J_c(\theta)$. The optimal solution $(\theta^*, \lambda^*)$ satisfies the Karush-Kuhn-Tucker (KKT) conditions:

**Stationarity:** $\nabla_\theta J_r(\theta^*) + \lambda^* \nabla_\theta J_c(\theta^*) = 0$,    **Primal feasibility:** $J_c(\theta^*) \leq 0$,
**Dual feasibility:** $\lambda^* \geq 0$,    **Complementary slackness:** $\lambda^* J_c(\theta^*) = 0$.

Generally, the optimal solution $(\theta^*, \lambda^*)$ can be found by solving the following min-max problem using the dual ascent-descent method: $\theta^*, \lambda^* \leftarrow \max_\lambda \min_\theta \mathcal{L}(\theta, \lambda)$.

## 4 METHOD

### 4.1 WHEN REWARD AND SAFETY CONFLICT

Generally, the gradients from the reward and safety objectives, denoted as $g_r = \nabla_\theta J_r(\theta)$ and $g_c = \nabla_\theta J_c(\theta)$, do not always point in the same direction. In fact, they often pull the policy in relatively opposite ways. A natural indicator of such disagreement is the inner product between the two gradients. If the inner product $\langle g_r, g_c \rangle$ is negative, which means the angle between them is obtuse, they are in conflict.

The qualitatively geometric interpretation is easy to understand. When the inner product is negative, if we step in the direction of $g_r$, the reward may improve, but safety will likely worsen, and vice versa. When the inner product is positive, the two gradients are more aligned, and it's possible to improve both objectives at the same time with a reasonable step size.

To better quantitatively understand the consequence of this conflict, we can consider how each objective changes after one step updating. Suppose the actual update direction is $h$, and we execute one step gradient descent as $\theta \leftarrow \theta - \beta h$ with learning rate $\beta > 0$. Then the first-order improvement $\Delta$ in each objective $J_i$ is roughly proportional to $\langle g_i, h \rangle$ since $\Delta = J_i(\theta - \beta h) - J_i(\theta)$ and $J_i(\theta - \beta h) \approx J_i(\theta) - \beta h^\mathsf{T} \nabla_\theta J_i(\theta) = J_i(\theta) - \beta \langle h, g_i \rangle$. That is, if the update direction h has a negative inner product with an objective's gradient (i.e., $\langle h, g_i \rangle < 0$), we are actually making things worse for that objective to be minimized at this iteration, since the change would be $\Delta \approx -\beta \langle h, g_i \rangle > 0$.

In this work, we focus on alleviating this gradient conflict issue by calculating a harmonic gradient (HG) for the policy update at each iteration, which minimizes the conflict with both $g_r$ and $g_c$.

## 4.2 HARMONIC LAGRANGIAN WITH MINIMAL CONFLICT

To alleviate gradient conflict and encourage improvement in both reward and safety performance, we regard the inner product value as the optimization target, and design a minimax optimization problem at each iteration to solve a harmonic gradient $h \in \mathbb{R}^n$ for policy updating. Since existing training methods all use small batch sampling to calculate the gradient and the magnitude of the gradient also needs to be restricted, we design a trust region constraint in a closed ball form, whose center is a nominal gradient i.e., $\hat{g} = (g_r + \lambda g_c)$, and whose radius depends on the length of the nominal gradient $\hat{g}$. The full formulation is given by

$$\min_{h \in \mathbb{R}^n} \max_{i \in \{r,c\}} -\langle g_i, h \rangle$$
$$\text{s. t. } \|h - \hat{g}\| \leq \rho\|\hat{g}\|, \tag{1}$$

where $n$ is the number of policy parameters, $\rho \in (0,1)$ is called harmonic constant that controls the range of the trust region closed ball.

This minimax problem Eq. (1) has two iteration loops: the inner loop is to identify the worse-case objective, and the outer loop is to optimize its inner product within the trust region. We give a geometric interpretation in Figure 2: two black arrows represent $g_r$ and $g_c$, blue arrow represents $\hat{g}$, yellow circle represents the trust region and red arrow denotes $h$. It can be seen that there is a conflict between $\hat{g}$ and $g_r$, but our $h$ neither conflicts with $g_r$ nor $g_c$. In addition, we can also intuitively see the effect of $\rho$ on the trust region. When $\rho > 0$, it controls how much room there is to adjust $h$ for mitigating gradient conflicts, and a value of $\rho = 0$ implies that $h$ precisely equals $\hat{g}$.

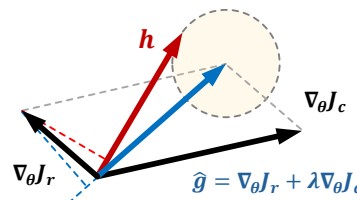

Figure 2: Geometric interpretation of harmonic Lagrangian, where the red arrow is the harmonic gradient.

## 4.3 HARMONIC CONSTRAINED REINFORCEMENT LEARNING

HCRL is a framework that incorporates harmonic gradient into CRL algorithms to balance reward and safety objectives. This is accomplished by first computing two separate policy gradients—one for reward and one for safety—and then deriving their harmonic gradient. The harmonic gradient is solved with minimal conflict, ensuring smooth policy updates at each RL iteration.

Although the form of Eq. (1) appears simple, it is quite challenging to solve for several reasons: (1) It involves two iteration loops, requiring alternating optimization of the inner and outer loops, which are prone to oscillations and make convergence difficult (Ren et al., 2023). (2) The inner loop is a discrete optimization problem, typically necessitating the traversal of all possible options one by one (Krause et al., 2013). (3) The outer loop has a high variable dimension, equal to the number of policy parameters, which can reach millions in the case of a deep neural network. (4) The trust region constraint is an inequality constraint on $h$, which is also high-dimensional. Consequently, existing solvers are not capable of directly solving this problem.

**Practical implementation.** To address these challenges, we apply a series of equivalent problem transformations to enable practical solving with significantly improved computational efficiency. The resulting solution is summarized in the following theorem.

**Theorem 4.1** (Harmonic Gradient). *The solution to* Eq. (1) *is given by*

$$h = \hat{g} + \rho \frac{\|\hat{g}\|}{\|d(\alpha^*)\|} d(\alpha^*), \tag{2}$$

*where $d(\alpha)$ is the linear combination of $g_r$ and $g_c$ as $d(\alpha) = \alpha g_r + (1 - \alpha)g_c, \alpha \in [0,1]$, and $\alpha^*$ can be numerically determined by solving*

$$\alpha^* = \arg \max_{\alpha \in [0,1]} \langle d(\alpha), \hat{g} \rangle + \rho\|\hat{g}\|\|d(\alpha)\|. \tag{3}$$

*Proof.* See Appendix B. □

This theorem simplifies the ***original high-dimensional constrained two-loop minimax problem*** in Eq. (1) as a ***one-dimensional unconstrained one-loop problem*** in Eq. (3), enables efficiently solving the harmonic gradient at each RL iteration. The complete pseudocode is presented in Algorithm 1.

We also provide theoretical analysis in Appendix C to claim that incorporating HG into Lagrangian-based methods does not compromise their original guarantees, including convergence and optimality.

---

**Algorithm 1 HCRL**: Harmonic Constrained Reinforcement Learning

---

**Input:** Harmonic constant $\rho_0 \in [0, 1)$; learning rates $\beta_1, \beta_2, \beta_3 > 0$ for multiplier, policy, values
**Initialize:** Parameters $\theta_0, \omega_0, \phi_0, \lambda_0$ for policy, value, cost value, multiplier
 1: **for** each iteration $k$ **do**
 2:     Compute reward gradient $g_r$ and cost gradient $g_c$
 3:     // Multiplier Update
 4:     Compute multiplier gradient: $\nabla_\lambda \mathcal{L}$
 5:     Update Lagrange multiplier: $\lambda_{k+1} = \max\left(0, \ \lambda_k + \beta_1 \nabla_\lambda \mathcal{L}\right)$
 6:     // Actor Update
 7:     Compute harmonic policy gradient $h_k$ (see Eq. (2))        ▷ Calculate HG for policy update
 8:     Update policy: $\theta_{k+1} = \theta_k - \beta_2 h_k$
 9:     // Critic Update
10:     Compute gradients: $g_\omega$ for value, $g_\phi$ for cost value
11:     Update values: $\omega_{k+1} = \omega_k - \beta_3 g_\omega, \quad \phi_{k+1} = \phi_k - \beta_3 g_\phi$
12: **end for**
13: **return** final parameters $\theta, \omega, \phi, \lambda$

---

## 5 EXPERIMENTS

We seek to answer the following three questions through our experiments:

**Q1.** Can HCRL enhance the stability of convergence in both convex and non-convex planar constrained optimization tasks?

**Q2.** Can HCRL achieve faster convergence, in a quantitative sense, compared to other Lagrangian-based algorithms across varying initializations?

**Q3.** Can HCRL outperform or match the performance of existing state-of-the-art CRL algorithms in complex safe control tasks?

### 5.1 PLANAR CONSTRAINED OPTIMIZATION

We first employ three planar constrained optimization problem, one convex and two non-convex, to assess the performance of our designed harmonic Lagrangian. More descriptions are in Appendix F.1.



**Convex Problem**  **Non-Convex Problem 1**



$$\min_{x,y} (x - 2)^2 + (y - 3)^2, \qquad \min_{x,y} (x - 2)^4 - 3(x - 2)^2 + (y - 1)^4 - (y - 1)^2,$$
$$\text{s.t. } x^2 + y - 4 \le 0. \qquad\qquad \text{s.t. } x^2 - \cos y - 1 \le 0.$$



**Non-Convex Problem 2**



$$\min_{x,y} (x - 2)^4 - 3(x - 2)^2 + (y - 1)^4 - 2(y - 1)^2 + 2\sin(3x)\cos(3y),$$
$$\text{s.t. } x^2 + 0.5\sin(2y) - 1.2\cos(1.5x) - 1 = 0.$$

We consider two baselines: the naive Lagrangian (Ray et al., 2019) and PID Lagrangian methods (Stooke et al., 2020). The PID Lagrangian method employs a PID controller to adjust the multiplier, which empirically improves stability. We set the gain coefficients as $K_p = 1.0$, $K_I = 0.005$, and $K_D = 0.001$, following the settings from the original reference (Stooke et al., 2020). We visualize several typical iteration processes in Figure 3, where Lag, PID-Lag, H-Lag denote naive

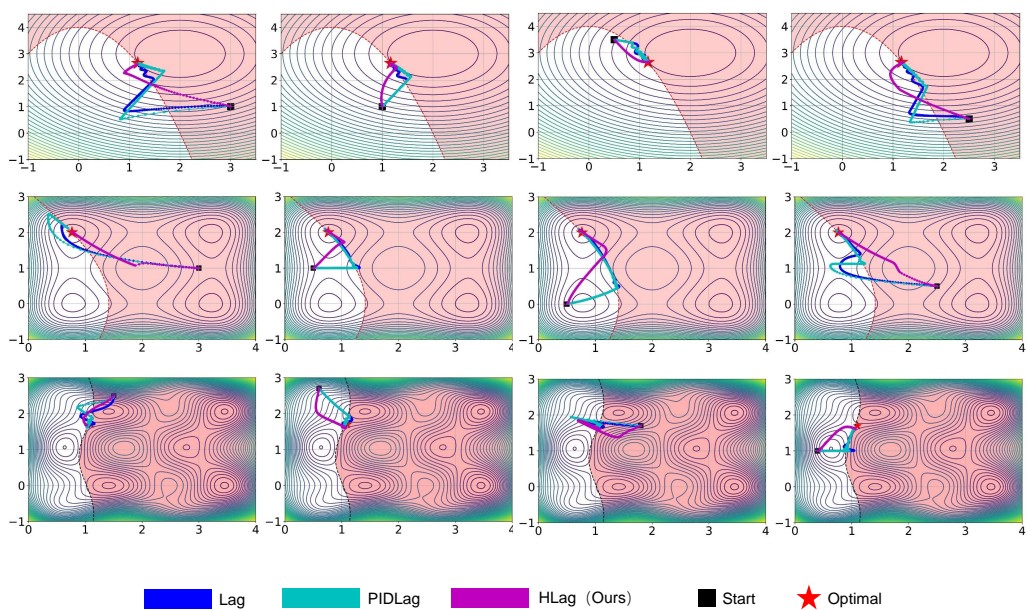

Figure 3: Typical iteration process comparisons on the convex and non-convex problems

Lagrangian, PID Lagrangian, and the harmonic Lagrangian of our HCRL, respectively. It can be seen that the PID-Lag method performing better than the naive Lag, but our HCRL (HLag) achieves the most stable and fastest convergence, effectively avoiding oscillations near constraint boundaries, answering **Q1**.

Table 1: Performance comparison. ↑ indicates higher is better, ↓ indicates lower is better.

| Metric | Problem | Lag | PIDLag | HLag (ours) |
|---|---|---|---|---|
| | Convex | **100%** | **100%** | **100%** |
| Success ↑ | Non-convex1 | 80% | 89% | **98%** |
| | Non-convex2 | 75% | 90% | **95%** |
| | Convex | 228.40 | 195.63 | **102.53** |
| Iteration ↓ | Non-convex1 | 909.70 | 785.90 | **550.74** |
| | Non-convex2 | 1050.20 | 850.30 | **600.40** |

To quantitatively evaluate performance and convergence speed, we randomly sample 100 initial points within the plane region and iteratively search for the optimal solution from each point. Table 1 reports the average success rate and the number of iterations required for convergence for all three methods. While all methods achieve a 100% success rate on the convex problem, our HCRL significantly outperforms the two baselines on the non-convex problems, achieving high success rates of 98% and 95%, respectively. Moreover, HCRL consistently requires the fewest iterations across all three problems, demonstrating both higher reliability and improved efficiency, answering **Q2**.

## 5.2 SAFETY GYMNASIUM BENCHMARK

**Benchmark.** We evaluate our method on 10 complex safe control tasks from the Safety Gymnasium (Ji et al., 2023) to benchmark reward and safety performance. These tasks include 4 navigation (reach goal and avoid collision) and 6 locomotion (move forward and avoid overspeed) tasks. The snapshots of these tasks are shown in Figure 4 and more detailed descriptions can be found in Appendix F.2.

**Our methods.** We build on two state-of-the-art model-free RL backbones: the off-policy algorithm DSAC (Duan et al., 2024) and the on-policy algorithm PPO (Schulman et al., 2017). By incorporating our proposed HCRL, we derive two safe-oriented variants, namely DSAC-HLag and PPO-HLag.

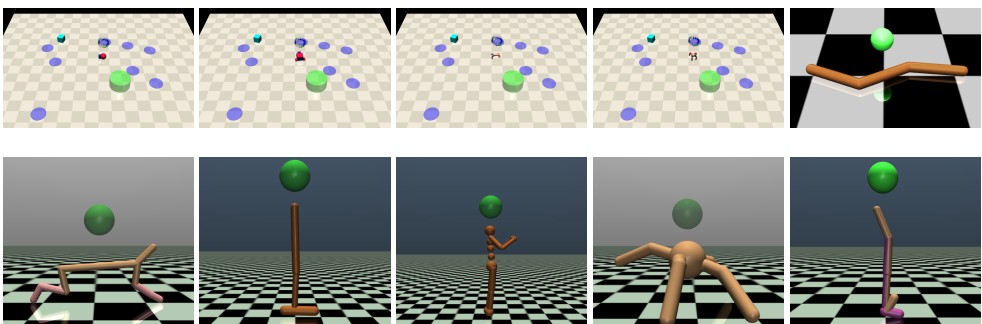

Figure 4: Snapshots of 10 Safety Gymnasium tasks.

**Baselines.** For comparison, we evaluate our two HG-equipped methods against eight baselines: three off-policy methods: naive DSAC (Duan et al., 2024), DSAC-Lag (Ray et al., 2019), and DSAC-PIDLag (Stooke et al., 2020), and five on-policy methods: PPO (Schulman et al., 2017), CPO (Achiam et al., 2017), RCPO (Tessler et al., 2019), PPO-Lag (Ray et al., 2019), and IPO (Liu et al., 2020a). Implementation details are provided in Appendix G.

**Main results.** For constrained control tasks, an effective policy must achieve both high rewards and low costs. Therefore, We consider two metrics simultaneously: total average reward (TAR) and total average cost (Cost), and evaluate all methods across 10 benchmark tasks and report the averaged values over 5 runs in Table 2. As seen, our HG-equipped methods (in red) outperform their respective baselines in either TAR, Cost, or both. In particular, DSAC-HLag achieves higher TAR than all safe RL baselines (in blue) while maintaining a competitive Cost, demonstrating a more favorable balance between reward and safety. Detailed results and training curves are presented in Appendix A.

**TAR-Cost evaluation.** To provide a more intuitive comparison, we plot TAR and Cost averaged on the 10 tasks in a two-dimensional plane as shown in Figure 5: the vertical axis denotes TAR (higher is better), while the horizontal axis denotes Cost (smaller is better, so a shift to the right indicates a safer outcome). Collectively, methods closer to the top-right corner achieve a better trade-off between reward maximization and safety. We highlight different categories of baselines with colors for clarity: unsafe-oriented general methods (green), safe-oriented methods (blue), and our proposed methods (red). As shown, both DSAC-HLag and PPO-HLag consistently occupy the top-right corner, demonstrating superior overall performance. Importantly, compared with their respective backbone algorithms DSAC-Lag and PPO-Lag, our methods achieve improvements in both reward and cost simultaneously. This highlights the effectiveness of our proposed HG in effectively guiding policies towards feasible and optimal solutions, answering **Q3**.

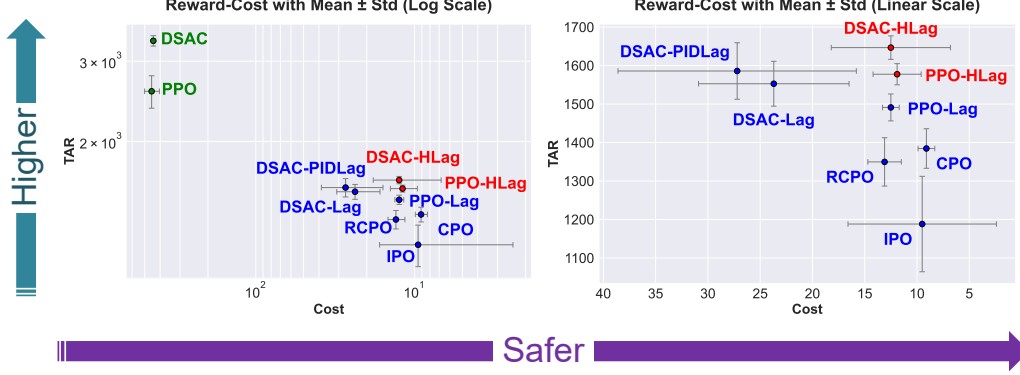

Figure 5: **Visualization of TAR-Cost evaluation.** Note that the x-axis (Cost) is reversed: moving to the right indicates lower cost, which corresponds to higher safety. The y-axis (TAR) is standard, where moving upward indicates higher performance.

Table 2: Comparison of TAR and Cost across different algorithms. TAR $\uparrow$ indicates higher is better, Cost $\downarrow$ indicates lower is better. The results of our HG-augmented method are highlighted in bold.

| Paradigm | Category | Algorithm | TAR $\uparrow$ | Cost $\downarrow$ |
|---|---|---|---|---|
| Off-policy | Unsafe-oriented | DSAC | $3324.5 \pm 87.3$ | $441.7 \pm 7.2$ |
| | Safe-oriented | DSAC-Lag | $1552.8 \pm 58.3$ | $23.7 \pm 7.0$ |
| | | DSAC-PIDLag | $1585.9 \pm 73.4$ | $27.2 \pm 11.4$ |
| | | DSAC-HLag | $\mathbf{1646.7 \pm 30.5}$ | $\mathbf{12.5 \pm 5.7}$ |
| On-policy | Unsafe-oriented | PPO | $2577.1 \pm 210.5$ | $452.8 \pm 48.4$ |
| | Safe-oriented | PPO-Lag | $1491.2 \pm 34.9$ | $12.5 \pm 0.8$ |
| | | RCPO | $1349.7 \pm 62.8$ | $13.1 \pm 1.6$ |
| | | IPO | $1188.2 \pm 124.0$ | $9.5 \pm 7.1$ |
| | | CPO | $1384.5 \pm 51.4$ | $9.1 \pm 0.8$ |
| | | PPO-HLag | $\mathbf{1577.5 \pm 27.5}$ | $\mathbf{11.9 \pm 2.3}$ |

*\* Each value is the average result across the 10 safety gymnasium tasks.*
*\* The results of each task are the average of 5 runs.*

## 5.3 ABLATION STUDY

To verify the effectiveness of our HCRL under different situations, we further conduct several ablations and time-efficiency analysis.

**(1) Ablation on trust region size controlled by $\rho$.** The trust region size plays a crucial role in balancing convergence speed and update smoothness. We further investigated the influence of the trust region size on the Walker2d-Vel task using DSAC-HLag to assess its impact on both reward and safety. The results are listed in Table 3. Overall, our method demonstrates robustness to the choice of $\rho$, achieving consistent TAR and steadily decreasing cost across a wide range

Table 3: Performance under varying trust region sizes.

| Trust region size | TAR $\uparrow$ | Cost $\downarrow$ |
|---|---|---|
| $\rho = 0.1$ | $2921.7 \pm 48.4$ | $24.2 \pm 6.3$ |
| $\rho = 0.3$ | $2919.1 \pm 46.0$ | $16.3 \pm 4.6$ |
| $\rho = 0.5$ | $2912.5 \pm 49.1$ | $10.8 \pm 3.1$ |
| $\rho = 0.7$ | $2908.6 \pm 48.9$ | $9.3 \pm 2.8$ |
| $\rho = 0.9$ (default) | $2903.9 \pm 48.8$ | $8.1 \pm 2.6$ |

$(0.5 - 0.9)$. Notably, when the $\rho$ is very small $(0.1 - 0.3)$, safety performance degrades significantly, accompanied by only marginal TAR improvement, likely due to overly restricted policy updates. In contrast, larger $\rho$ enable a broader range of harmonic gradient search, which alleviates gradient conflicts and stabilizes training, ultimately yielding more balanced and feasible solutions.

Table 4: Ablation study on the initial value and learning rate of multiplier. Underlined are the default.

| | TAR $\uparrow$ | Cost $\downarrow$ | TAR $\uparrow$ | Cost $\downarrow$ | TAR $\uparrow$ | Cost $\downarrow$ |
|---|---|---|---|---|---|---|
| | $\lambda_0 = 0.1$ | | $\lambda_0 = 1.0$ | | $\lambda_0 = 10.0$ | |
| Walker2dVel | $\mathbf{2903.9 \pm 48.8}$ | $8.1 \pm 2.6$ | $2900.8 \pm 34.5$ | $13.7 \pm 4.2$ | $2901.5 \pm 71.9$ | $\mathbf{7.7 \pm 4.1}$ |
| | $\beta_1 = 1e\text{-}6$ | | $\underline{\beta_1 = 1e\text{-}5}$ | | $\beta_1 = 1e\text{-}4$ | |
| Walker2dVel | $\mathbf{2942.9 \pm 45.2}$ | $13.0 \pm 6.3$ | $2903.9 \pm 48.8$ | $8.1 \pm 2.6$ | $2902.1 \pm 104.1$ | $\mathbf{7.5 \pm 3.7}$ |

**(2) Ablation on initial value and learning rate of multiplier.** We further conduct ablation studies on the initial value and learning rate of the Lagrangian multiplier using DSAC-HLag on the Walker2d-Vel task. In these experiments, our default setting is an initial multiplier of 0.1 and a learning

rate of $1 \times 10^{-5}$. Additionally, we explored other configurations including initial values of 1 and 10, and initial learning rates of $1 \times 10^{-4}$ and $1 \times 10^{-6}$ (note that we use the learning rate decay technique and the final learning rate is $1 \times 10^{-6}$). As summarized in Table 4, we observe that smaller initial multipliers and learning rates tend to yield higher TAR values, indicating improved reward performance, while larger multipliers and learning rates result in lower Cost values, reflecting stronger safety enforcement. Across all configurations, our method consistently outperforms the baseline algorithms, demonstrating the robustness of our framework and its ability to maintain a favorable balance between reward maximization and safety preservation.

**(3) Ablation on cost limit.** We further investigate the effect of varying the cost limit using DSAC-HLag on the Walker2d-Vel task, with results summarized in Table 5. We consider two representative settings: a strict constraint (limit = 0) and a relaxed constraint (limit = 20). Across both cases, our HCRL algorithm consistently outperforms the PIDLag baseline. Notably, when the cost limit is relaxed to 20, HCRL still achieves significantly higher TAR while still keeping the Cost lower or comparable relative to the allowed budget. These findings highlight that HCRL not only enforces stricter safety requirements under tight constraints, but also flexibly adapts to looser safety budgets, achieving a more favorable trade-off between reward maximization and constraint satisfaction.

Table 5: Ablation study on the cost limit (0 or 20). w/o cost limit is the default setting we used above.

| | TAR ↑ | Cost ↓ | TAR ↑ | Cost ↓ |
|---|---|---|---|---|
| | PIDLag (w/o cost limit 20) | | PIDLag (w cost limit 20) | |
| Walker2dVel | $2942.0 \pm 47.4$ | $26.5 \pm 5.4$ | $2928.6 \pm 44.8$ | $27.0 \pm 6.8$ |
| | HLag (w/o cost limit) | | HLag (w cost limit) | |
| Walker2dVel | $2903.9 \pm 48.8$ | $\mathbf{8.1 \pm 2.6}$ | $\mathbf{3029.9 \pm 40.2}$ | $17.5 \pm 4.6$ |

**(4) Training time comparison.** Our HCRL method introduces an additional step to compute the HG for policy update. To examine its impact on the time-efficiency, we measure the training time per 1k steps on the PointGoal and AntGoal tasks. In PointGoal, the recorded times for the 3 off-policy algorithms were [182s, 187s, 239s], while in AntGoal they were [278s, 284s, 360s]. These results are visualized in Figure 6. All experiments are conducted using an AMD Threadripper 3960X CPU and an NVIDIA RTX 3090Ti GPU. Overall, our method incurs an additional 29%–32% computational burden. We argue that this overhead is acceptable given the substantial performance improvements achieved by HCRL.

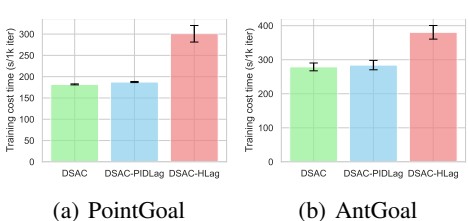

(a) PointGoal      (b) AntGoal

Figure 6: Training time-efficiency comparison. Green, blue and red correspond to DSAC, DSAC-Lag and DSAC-HLag, respectively.

## 6 CONCLUSION

This paper introduces the Harmonic Constrained Reinforcement Learning (HCRL) framework, a novel approach for optimally balancing reward maximization and safety satisfaction at the gradient level. By solving for a harmonic gradient (HG) for policy update at each RL iteration, HCRL mitigates conflicts between the reward and safety objectives and consequently enhances both the stability and efficiency of the learning process towards optimal feasible solutions. Empirical results on 10 Safety Gymnasium tasks demonstrate that our HCRL consistently outperforms or matches the performance of state-of-the-art constrained RL algorithms in terms of both reward and safety.

To conclude, our proposed HCRL is orthogonal to existing constrained optimization techniques. It is not a standalone algorithm but rather a general enhancement that can be integrated into existing constrained RL algorithms (whether off-policy or on-policy) to more effectively search the optimal and feasible solutions. We believe that this work opens up a new direction towards developing more robust and reliable constrained RL agents.

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

## A  SUPPLEMENTARY RESULTS & TRAINING CURVES

### A.1  RESULTS OF OFF-POLICY METHODS

Table 6: Numerical results in 10 Safety Gymnasium tasks (off policy). For the right 3 CRL algorithms (The backbone DSAC is not included), the highest TAR and lowest Cost in each row are **highlighted**.

| Task | DSAC | | DSAC-Lag | | DSAC-PIDLag | | DSAC-HLag (ours) | |
|---|---|---|---|---|---|---|---|---|
| | TAR ↑ | Cost ↓ | TAR ↑ | Cost ↓ | TAR ↑ | Cost ↓ | TAR ↑ | Cost ↓ |
| PointGoal | $28.9 \pm 0.5$ | $7.9 \pm 3.9$ | $26.8 \pm 0.3$ | $2.8 \pm 2.0$ | $27.9 \pm 0.6$ | $4.3 \pm 1.1$ | $\mathbf{29.4 \pm 0.2}$ | $\mathbf{2.4 \pm 0.4}$ |
| CarGoal | $34.6 \pm 0.4$ | $12.3 \pm 3.5$ | $34.4 \pm 0.8$ | $11.4 \pm 2.1$ | $\mathbf{35.3 \pm 0.3}$ | $11.2 \pm 1.7$ | $35.0 \pm 0.2$ | $\mathbf{5.7 \pm 1.0}$ |
| DoggoGoal | $51.4 \pm 2.9$ | $11.1 \pm 3.8$ | $48.1 \pm 2.9$ | $10.8 \pm 1.8$ | $45.2 \pm 2.9$ | $11.6 \pm 0.9$ | $\mathbf{50.7 \pm 1.1}$ | $\mathbf{9.1 \pm 1.5}$ |
| AntGoal | $133.7 \pm 1.4$ | $15.9 \pm 3.0$ | $86.7 \pm 30.0$ | $13.8 \pm 2.5$ | $87.8 \pm 12.5$ | $11.5 \pm 1.7$ | $\mathbf{127.1 \pm 4.4}$ | $\mathbf{10.7 \pm 1.3}$ |
| SwimmerVel | $104.6 \pm 2.5$ | $121.9 \pm 4.9$ | $30.6 \pm 15.1$ | $42.2 \pm 2.6$ | $24.0 \pm 8.7$ | $31.2 \pm 8.3$ | $\mathbf{35.6 \pm 15.5}$ | $\mathbf{29.1 \pm 14.9}$ |
| HalfcheetahVel | $11595.9 \pm 262$ | $979.4 \pm 2.43$ | $2846.2 \pm 50.5$ | $\mathbf{0.4 \pm 0.2}$ | $\mathbf{2951.5 \pm 31.2}$ | $2.1 \pm 1.9$ | $2907.8 \pm 16.6$ | $0.5 \pm 0.2$ |
| HopperVel | $2261.5 \pm 120.0$ | $533.8 \pm 21.4$ | $1239.0 \pm 44.6$ | $68.8 \pm 9.7$ | $\mathbf{1286.6 \pm 34.1}$ | $111.8 \pm 40.4$ | $1243.8 \pm 52.3$ | $\mathbf{32.9 \pm 16.4}$ |
| HumanoidVel | $7602.3 \pm 166.3$ | $884.2 \pm 11.3$ | $5367.3 \pm 5.7$ | $57.0 \pm 25.8$ | $5842.1 \pm 200.8$ | $45.0 \pm 11.5$ | $\mathbf{6047.1 \pm 99.7}$ | $\mathbf{19.7 \pm 4.8}$ |
| AntVel | $6341.5 \pm 135.9$ | $948.6 \pm 8.5$ | $2901.5 \pm 205.9$ | $14.1 \pm 4.5$ | $2616.9 \pm 197.9$ | $17.2 \pm 3.4$ | $\mathbf{3086.9 \pm 35.6}$ | $\mathbf{6.6 \pm 2.6}$ |
| Walker2dVel | $5090.7 \pm 59.1$ | $901.9 \pm 22.3$ | $\mathbf{2947.1 \pm 38.8}$ | $16.1 \pm 8.5$ | $2942.0 \pm 47.4$ | $26.5 \pm 5.4$ | $2903.9 \pm 48.8$ | $\mathbf{8.1 \pm 2.4}$ |

### A.2  RESULTS OF ON-POLICY BASELINES

Table 7: Numerical results in 10 Safety Gymnasium tasks (on policy). For the 5 safe RL algorithms (The backbone PPO is not included), the highest TAR and lowest Cost in each row are **highlighted**.

| Task | PPO | | PPO-HLag (ours) | | RCPO | |
|---|---|---|---|---|---|---|
| | TAR ↑ | Cost ↓ | TAR ↑ | Cost ↓ | TAR ↑ | Cost ↓ |
| PointGoal | $26.4 \pm 1.9$ | $26.1 \pm 5.5$ | $21.4 \pm 1.4$ | $8.4 \pm 3.9$ | $\mathbf{23.7 \pm 0.6}$ | $7.0 \pm 0.7$ |
| CarGoal | $27.5 \pm 0.7$ | $14.4 \pm 2.5$ | $22.1 \pm 1.0$ | $8.4 \pm 2.6$ | $\mathbf{27.3 \pm 0.5}$ | $7.4 \pm 1.5$ |
| DoggoGoal | $15.9 \pm 0.8$ | $14.5 \pm 5.6$ | $\mathbf{8.7 \pm 0.3}$ | $11.7 \pm 4.4$ | $7.6 \pm 0.5$ | $8.4 \pm 4.2$ |
| AntGoal | $12.9 \pm 1.1$ | $8.3 \pm 1.8$ | $8.8 \pm 0.9$ | $4.7 \pm 1.3$ | $13.1 \pm 1.2$ | $5.9 \pm 1.9$ |
| SwimmerVel | $108.5 \pm 7.4$ | $982.2 \pm 16.8$ | $\mathbf{36.9 \pm 0.2}$ | $45.6 \pm 0.2$ | $26.0 \pm 3.4$ | $44.9 \pm 3.8$ |
| HalfcheetahVel | $6934.8 \pm 544.4$ | $975.9 \pm 3.1$ | $2540.0 \pm 50.0$ | $13.9 \pm 0.7$ | $2396.7 \pm 184.8$ | $\mathbf{10.7 \pm 0.2}$ |
| HopperVel | $2094.2 \pm 188.9$ | $469.1 \pm 55.4$ | $\mathbf{1343.9 \pm 16.0}$ | $4.7 \pm 3.8$ | $602.9 \pm 109.2$ | $0.9 \pm 0.2$ |
| HumanoidVel | $6539.3 \pm 338.2$ | $264.5 \pm 166.5$ | $\mathbf{6342.2 \pm 58.4}$ | $10.2 \pm 7.7$ | $4986.3 \pm 111.1$ | $33.3 \pm 4.2$ |
| AntVel | $5093.7 \pm 289.6$ | $885.7 \pm 16.2$ | $2989.0 \pm 70.0$ | $9.9 \pm 0.2$ | $2968.8 \pm 99.6$ | $9.6 \pm 0.1$ |
| Walker2dVel | $4917.5 \pm 520.9$ | $887.5 \pm 48.8$ | $2461.8 \pm 52.8$ | $\mathbf{1.1 \pm 0.2}$ | $2444.6 \pm 50.5$ | $2.6 \pm 1.5$ |

| Task | PPO-Lag | | IPO | | CPO | |
|---|---|---|---|---|---|---|
| | TAR ↑ | Cost ↓ | TAR ↑ | Cost ↓ | TAR ↑ | Cost ↓ |
| PointGoal | $22.3 \pm 1.0$ | $7.6 \pm 1.8$ | $19.1 \pm 2.8$ | $6.1 \pm 1.2$ | $22.6 \pm 0.6$ | $\mathbf{5.9 \pm 1.0}$ |
| CarGoal | $24.9 \pm 0.6$ | $7.9 \pm 1.1$ | $22.8 \pm 0.9$ | $\mathbf{6.5 \pm 1.0}$ | $27.0 \pm 1.1$ | $7.2 \pm 2.3$ |
| DoggoGoal | $8.1 \pm 0.4$ | $10.1 \pm 2.3$ | $3.1 \pm 0.6$ | $\mathbf{4.3 \pm 4.0}$ | $5.2 \pm 0.2$ | $5.8 \pm 2.4$ |
| AntGoal | $11.4 \pm 1.0$ | $5.2 \pm 1.2$ | $3.8 \pm 0.3$ | $\mathbf{2.7 \pm 0.9}$ | $\mathbf{14.5 \pm 0.9}$ | $4.9 \pm 0.9$ |
| SwimmerVel | $31.5 \pm 1.5$ | $45.2 \pm 1.1$ | $8.1 \pm 17.4$ | $49.9 \pm 11.0$ | $23.8 \pm 4.9$ | $\mathbf{33.6 \pm 1.1}$ |
| HalfcheetahVel | $\mathbf{2699.0 \pm 1.8}$ | $12.1 \pm 0.5$ | $2500.1 \pm 223.4$ | $13.6 \pm 23.7$ | $2559.6 \pm 27.0$ | $11.0 \pm 0.5$ |
| HopperVel | $985.0 \pm 80.0$ | $2.7 \pm 1.2$ | $616.7 \pm 223.5$ | $\mathbf{0.5 \pm 0.2}$ | $559.7 \pm 46.0$ | $0.8 \pm 0.2$ |
| HumanoidVel | $5660.0 \pm 100.0$ | $22.1 \pm 3.0$ | $4530.3 \pm 220.9$ | $\mathbf{4.0 \pm 1.1}$ | $5258.1 \pm 134.3$ | $13.1 \pm 2.1$ |
| AntVel | $\mathbf{3020.0 \pm 16.6}$ | $9.8 \pm 0.2$ | $2058.2 \pm 353.5$ | $\mathbf{5.0 \pm 2.0}$ | $2911.8 \pm 134.6$ | $7.4 \pm 0.2$ |
| Walker2dVel | $2450.0 \pm 30.0$ | $1.9 \pm 0.5$ | $2119.4 \pm 84.6$ | $2.5 \pm 0.3$ | $\mathbf{2462.3 \pm 72.3}$ | $1.5 \pm 0.3$ |

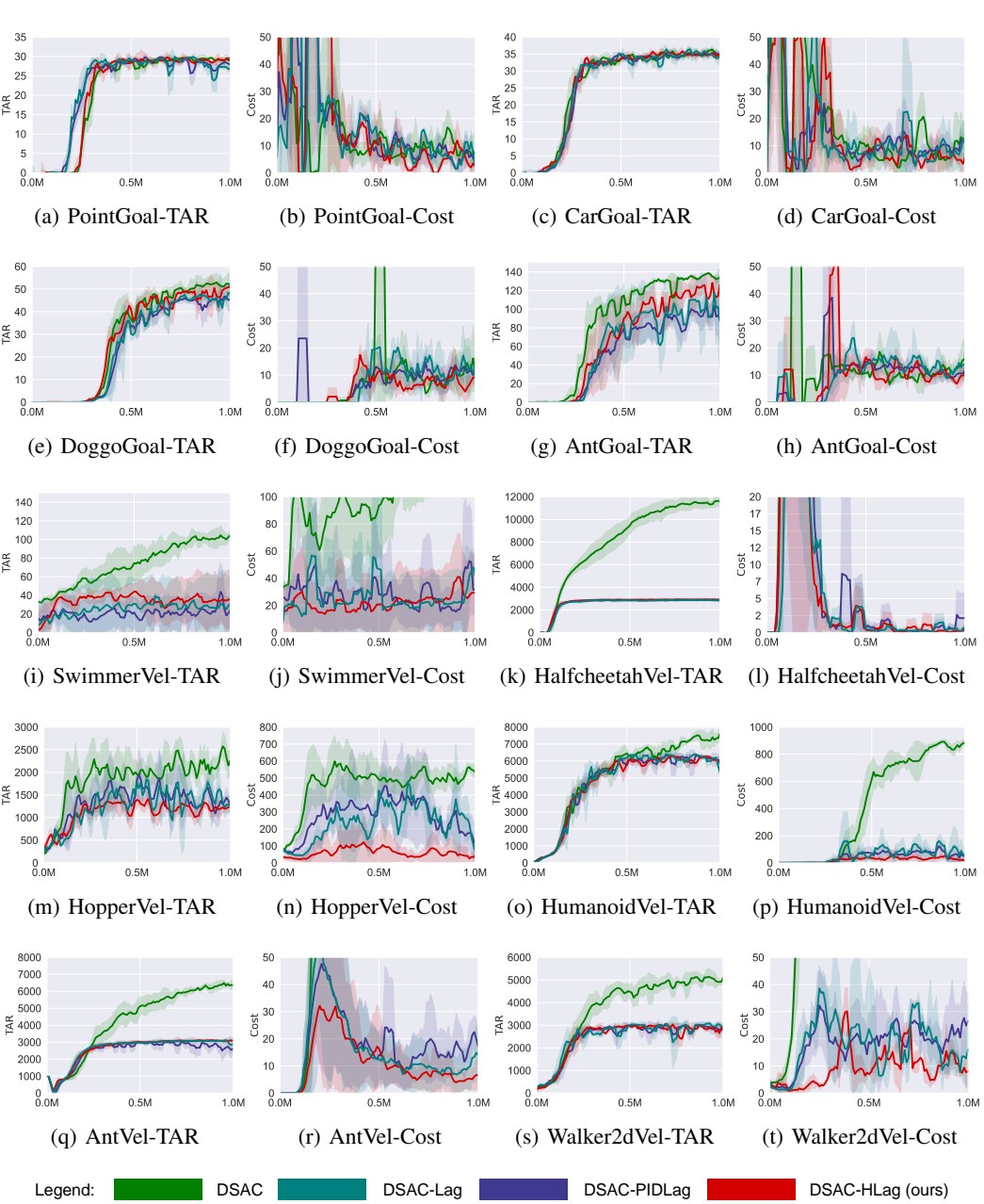

Figure 7: Training curves of 10 Safety Gymnasium tasks over 5 runs. Here the **TAR** means the total average reward return, and the **Cost** means the total average cost return.

## B  PRACTICAL IMPLEMENTATIONS OF HCRL

### B.1  EQUIVALENTLY TRANSFORM THE INNER LOOP AS A LINEAR PROGRAMMING

We first observe that the inner loop can be equivalently transformed as a convex linear programming problem with a continuous variable $\alpha \in [0, 1]$ as

$$\max_{\alpha \in [0,1]} -\langle d(\alpha), h \rangle, \tag{4}$$

where

$$d(\alpha) = \alpha g_r + (1 - \alpha)g_c. \tag{5}$$

The reason for the equivalence of this transformation is that the optimal solution of a linear programming can only be obtained at the vertices of the definition domain, which are $d(0) = g_c$ and $d(1) = g_r$ (Dantzig, 2016).

### B.2  APPLY LAGRANGIAN DUAL TRANSFORMATION TO ADDRESS TRUST REGION CONSTRAINT

To acquire a better convexity, we apply a common squaring technique in convex optimization (Boyd & Vandenberghe, 2004): equivalently replacing the original inequality constraint $\|h - \hat{g}\| \le \rho\|\hat{g}\|$ with

$$\|h - \hat{g}\|^2 \le \rho^2\|\hat{g}\|^2. \tag{6}$$

Consequently, the new problem becomes

$$\min_{h \in \mathbb{R}^n} \max_{\alpha \in [0,1]} -\langle d(\alpha), h \rangle$$
$$\text{s.t. } \|h - \hat{g}\|^2 \le \rho^2\|\hat{g}\|^2. \tag{7}$$

Next, we introduce a Lagrangian dual variable $\varphi \ge 0$ to address the trust region constraint, which is given by

$$\max_{\varphi \ge 0} \min_{h \in \mathbb{R}^n} \max_{\alpha \in [0,1]} -\langle d(\alpha), h \rangle + \frac{\varphi}{2}(\|h - \hat{g}\|^2 - \rho^2\|\hat{g}\|^2). \tag{8}$$

Prior to solve this dual problem, we first introduce two lemmas to analyze its convex and duality property.

**Lemma B.1** (Dual problem and convexity (Boyd & Vandenberghe, 2004)). *The dual problem is always a convex optimization problem, but it does not necessarily achieve the same optimal value.*

**Lemma B.2** (Slater condition and strong duality (Boyd & Vandenberghe, 2004)). *A convex problem satisfies Slater's condition if there exists a feasible point in the domain of the primal problem such that all the inequality constraints are strictly satisfied. When the Slater's condition holds (and the primal problem is convex), strong duality holds.*

Lemma B.1 directly confirms that Eq. (8) is a convex problem. Furthermore, by applying Lemma B.2, we can easily verify that Eq. (8) satisfies strong duality. The primal problem Eq. (7) is convex since it has a linear objective and convex constraint. Then we continue to prove that there exists a feasible point in the domain of the primal problem. One can take $h$ as $\frac{\rho}{2} \cdot \hat{g} + \hat{g}$, and the less than sign in the inequality constraint holds since $\|h - \hat{g}\|^2 = \left\|\frac{\rho}{2} \cdot \hat{g} + \hat{g} - \hat{g}\right\|^2 = \left\|\frac{\rho}{2} \cdot \hat{g}\right\|^2 \le \rho^2\|\hat{g}\|^2$. Therefore, according to Lemma B.2, the strong duality applies, indicating the the solution of the dual problem Eq. (8) is identical to the primal problem Eq. (1) and this dual transformation is also an equivalent problem transformation.

### B.3  PRACTICAL APPROACH FOR SOLVING HARMONIC GRADIENT

We first provide the proof of Theorem 4.1.

*Proof.* Through two equivalent problem transformations shown in Appendix B.1 and Appendix B.2, the resulting problem Eq. (8) has three iteration loops and also can not be solved easily with existing tools. Given that the convex property allows for the interchange of the $\min$ and $\max$ operators,

our core idea is to identify the special function forms with analytical solution so as to eliminate optimizable variables and reduce the number of iteration loops as many as possible.

After interchanging the operators, the resulting form is

$$\max_{\alpha\in[0,1]} \max_{\varphi\geq 0} \min_{h\in\mathbb{R}^n} -\langle d(\alpha), h\rangle + \frac{\varphi}{2}(\|h - \hat{g}\|^2 - \rho^2\|\hat{g}\|^2). \tag{9}$$

For the innermost minimization loop, this expression is in a quadratic function form with respect to $h$. Thus its optimal solution occurs at $\hat{g} + d(\alpha^*)/\varphi^*$, and we can replace $h$ with $\hat{g} + d(\alpha)/\varphi$ to remove this loop, which leads to

$$\max_{\alpha\in[0,1]} \max_{\varphi\geq 0} -\langle d(\alpha), \hat{g} + \frac{d(\alpha)}{\varphi}\rangle + \frac{\varphi}{2}(\left\|\frac{d(\alpha)}{\varphi}\right\|^2 - \rho^2\|\hat{g}\|^2). \tag{10}$$

Merge similar items, we have

$$\max_{\alpha\in[0,1]} \max_{\varphi\geq 0} -\langle d(\alpha), \hat{g}\rangle - \frac{1}{2\varphi}\|d(\alpha)\|^2 - \frac{\varphi\rho^2}{2}\|\hat{g}\|^2. \tag{11}$$

Then for the current inner maximization loop, this expression satisfies the check function form with respect to $\varphi$. Thus its optimal solution occurs at $\rho^{-1}\|d(\alpha^*)\|/\|\hat{g}\|$, and we can replace $\varphi$ with $\rho^{-1}\|d(\alpha)\|/\|\hat{g}\|$ to simplify out this loop, which leads to

$$\max_{\alpha\in[0,1]} \langle d(\alpha), \hat{g}\rangle + \rho\|\hat{g}\|\|d(\alpha)\|. \tag{12}$$

Finally, we end up with conducting optimization on the left maximization loop to solve

$$\alpha^* = \arg\max_{\alpha\in[0,1]} \langle d(\alpha), \hat{g}\rangle + \rho\|\hat{g}\|\|d(\alpha)\|. \tag{13}$$

With the conditions for replacement that

$$h = \hat{g} + d(\alpha^*)/\varphi^*, \quad \varphi = \rho^{-1}\|d(\alpha^*)\|/\|\hat{g}\|, \tag{14}$$

the harmonic gradient is naturally given by

$$h = \hat{g} + \frac{d(\alpha^*)}{\varphi^*} = \hat{g} + \rho\frac{\|\hat{g}\|}{\|d(\alpha^*)\|}d(\alpha^*). \tag{15}$$

This completes the proof. $\qquad\square$

Furthermore, we would like to emphasize the reduction in variable dimensionality resulting from our problem simplification with the following remark.

*Remark* B.3. The primal problem in Eq. (1) operates in the gradient space, whose dimensionality equals the number of policy parameters. This number can reach millions or even billions, which is highly impractical, as this problem must be solved at each iteration. **In contrast, the simplified problem in Eq. (13) only requires finding $\alpha^*$, a scalar, which significantly reduces the dimensionality and greatly improves computational efficiency.**

### B.4 ADAPTIVE DESIGN OF HARMONIC CONSTANT

We propose an adaptive design for the harmonic constant:

$$\rho_k = \rho_0 \sin\left(\frac{\phi_k}{2}\right), \tag{16}$$

where $\phi_k$ is the angle between $g_r$ and $g_c$. At each RL iteration, we can calculate the angle between the gradients to quantify the conflict degree, and $\rho$ is updated accordingly. As the conflict increases (i.e., the angle approaches 180 degrees), $\rho$ increases, expanding the trust region to allow a larger search space for mitigating the conflict more effectively.

## C MAIN THEORETICAL CONCLUSIONS

We first outline some necessary assumptions.

**Assumption C.1** (Lipschitz Continuity of Gradients). The gradients $g_r(\theta)$ and $g_c(\theta)$ are Lipschitz continuous, with positive constants $L_r$ and $L_c$ such that, for all $\theta, \theta' \in \mathbb{R}^n$:

$$\|g_r(\theta) - g_r(\theta')\| \le L_r\|\theta - \theta'\|, \quad \|g_c(\theta) - g_c(\theta')\| \le L_c\|\theta - \theta'\|. \tag{17}$$

**Assumption C.2** (Multi Time-Scale Step Sizes). The multiplier is on the slowest one $\{\beta_1(k)\}$, the policy update is on the intermediate time scale $\{\beta_2(k)\}$, and the critic update is on the fastest time scale $\{\beta_3(k)\}$. They satisfy: $\beta_1(k) < \beta_2(k) < \beta_3(k)$.

**Assumption C.3** (Boundedness of Policy Step Size). The policy step size $\{\beta_2(k)\}$ is bounded by $\beta_2(k) \le \frac{1}{L}$, where $L = L_r + \lambda_k L_c$.

We begin by proving the convergence of HCRL. Given that Assumption C.2 holds, the convergence of standard Lagrangian-based actor-critic RL has been well established (Yu et al., 2022). Our HCRL only modifies the policy update process, so we focus on proving that the policy will converge to a certain solution when the multiplier is frozen, denoted as $\lambda$. We first analyze the Lagrangian update bound.

**Theorem C.4** (Boundedness of Lagrangian Update). *With Assumption C.1 and Assumption C.3, The Lagrangian update at each step during iterative training satisfies*

$$\Delta_k = \mathcal{L}(\theta_{k+1}, \lambda) - \mathcal{L}(\theta_k, \lambda) \le -\frac{1}{2}\beta_2(1 - \rho_k^2)\|\hat{g}_k\|^2, \tag{18}$$

*where, with $\rho \in (0, 1)$, we have $\Delta_k \le 0$, indicating that the Lagrangian will decrease monotonically.*

*Proof.* See Appendix D.1. $\square$

Based on Theorem 4.1 and Theorem C.4, we can derive the convergence property as follows.

**Proposition C.5** (Convergence of Harmonic Gradient). *The harmonic gradient $h_k$ will converge to 0 as training progresses, i.e., $\lim_{k \to \infty} \|h_k\| = 0$.*

*Proof.* See Appendix D.2. $\square$

With Proposition C.5, we will prove the policy parameters sequence during the training satisfies the Cauchy sequence contraction mapping, thereby the convergence holds.

**Theorem C.6** (Convergence of HCRL). *The sequence $\theta_k$ satisfies the Cauchy sequence contraction theorem, for any $\epsilon > 0$, there exists an integer $N$ such that for all $m, n > N$:*

$$\|\theta_m - \theta_n\| \le \sum_{i=n}^{m-1} \|\theta_{i+1} - \theta_i\| = \sum_{i=n}^{m-1} \beta_2(i)\|h_i\| < \epsilon. \tag{19}$$

*Therefore, the policy parameters $\theta_k$ eventually converge to a certain solution $\tilde{\theta}$, i.e., $\lim_{k \to \infty} \theta_k = \tilde{\theta}$. Considering the whole well-established Lagrangian dual ascent-descent framework, the multiplier will also converge to a certain solution $\tilde{\lambda}$, i.e, $\lim_{k \to \infty} \lambda_k = \tilde{\lambda}$.*

*Proof.* See Appendix D.3. $\square$

Building on the convergence property, we next demonstrate that the converged solution satisfies the KKT conditions, which serve as the optimality criteria in the Lagrangian framework.

**Theorem C.7** (Optimality of HCRL). *The converged solution satisfies the KKT condition:*

***Stationarity:*** $\nabla_\theta J_r\left(\tilde{\theta}\right) + \tilde{\lambda}\nabla_\theta J_c\left(\tilde{\theta}\right) = 0$,    ***Primal feasibility:*** $J_c\left(\tilde{\theta}\right) \le 0$,

***Dual feasibility:*** $\tilde{\lambda} \ge 0$,                 ***Complementary slackness:*** $\tilde{\lambda}J_c\left(\tilde{\theta}\right) = 0$.

*Therefore, the converged $\tilde{\theta}$ serves as a candidate for the optimal solution $\theta^*$, indicating optimality.*

*Proof.* Refer to Appendix E.1 - E.4. $\square$

## D  PROOF OF CONVERGENCE

**Lemma D.1** (Triangle Inequality). *For any vectors $x, y \in \mathbb{R}^n$, the triangle inequality states that the norm of the sum of two vectors is less than or equal to the sum of their norms:*

$$\|x + y\| \leq \|x\| + \|y\|, \tag{20}$$

*where $\|\cdot\|$ denotes the standard Euclidean norm.*

**Lemma D.2** (Cauchy Sequence Contraction Theorem). *A sequence $\{a_n\}$ in a metric space $(X, d)$ is called a Cauchy sequence if for every $\epsilon > 0$, there exists a positive integer $N$ such that for all $m, n \geq N$, the following condition is satisfied:*

$$\|a_m - a_n\| < \epsilon. \tag{21}$$

*Furthermore, a sequence $\{a_n\}$ converges to a limit $a \in X$ if and only if $\{a_n\}$ is a Cauchy sequence and the space $X$ is complete, meaning every Cauchy sequence in $X$ converges to a limit in $X$.*

**Proposition D.3** (Boundedness of Lagrangian). *Assume $J_r(\theta)$ and $J_c(\theta)$ have lower bounds, denoted by $J_{r,min}$ and $J_{c,min}$, the Lagrangian $\mathcal{L}(\theta, \lambda)$ is lower bounded.*

*Proof.* Given that $J_r(\theta)$ and $J_c(\theta)$ are lower bounded, i.e.,

$$J_r(\theta) \geq J_{r,\min}, \quad J_c(\theta) \geq J_{c,\min}. \tag{22}$$

The Lagrange multiplier update rule is given by

$$\lambda_{k+1} = \max\left(0, \lambda_k + \beta_1 \nabla_\lambda \mathcal{L}(\theta_k, \lambda_k)\right). \tag{23}$$

This implies that the Lagrange multiplier $\lambda$ remains non-negative throughout the updates, i.e.,

$$\lambda \geq 0. \tag{24}$$

By substituting the lower bounds $J_{r,\min}$ and $J_{c,\min}$ into the expression of the Lagrangian $\mathcal{L}(\theta, \lambda) = J_r(\theta) + \lambda J_c(\theta)$, we obtain:

$$\begin{aligned}
\mathcal{L}(\theta, \lambda) &\geq J_{r,\min} + \lambda J_{c,\min} \\
&\geq J_{r,\min} + \lambda(-|J_{c,\min}|) \\
&\geq J_{r,\min} - \lambda|J_{c,\min}| \\
&= \mathcal{L}_{\min},
\end{aligned} \tag{25}$$

where $\mathcal{L}_{\min} = J_{r,\min} - \lambda|J_{c,\min}|$ represents the lower bound of the Lagrangian.

This completes the proof. $\qquad\square$

**Proposition D.4** (Boundedness of Inner Product). *The inner product $\hat{g}_k^T h_k$ is lower bounded.*

*Proof.* We start with the expression for the inner product:

$$\hat{g}_k^T h_k = \frac{1}{2} \left( \|\hat{g}_k\|^2 + \|h_k\|^2 - \|\hat{g}_k - h_k\|^2 \right). \tag{26}$$

According to the trust region constraint in Eq. (6), we have

$$\|\hat{g}_k - h_k\| \leq \rho_k \|\hat{g}_k\|, \tag{27}$$

where $\rho_k \in (0, 1)$ is the trust region parameter.

Substituting Eq. (27) into Eq. (26), we get

$$\begin{aligned}
\hat{g}_k^T h_k &\geq \frac{1}{2} \left( \|\hat{g}_k\|^2 + \|h_k\|^2 - \rho_k^2 \|\hat{g}_k\|^2 \right) \\
&= \frac{1}{2}(1 - \rho_k^2)\|\hat{g}_k\|^2 + \frac{1}{2}\|h_k\|^2.
\end{aligned} \tag{28}$$

This completes the proof. $\qquad\square$

## D.1 LAGRANGIAN UPDATE BOUND

*Proof.* Consider the difference in the Lagrangian before and after the $k$-th training iteration. We have:

$$\mathcal{L}\left(\theta_{k+1}, \lambda\right) - \mathcal{L}\left(\theta_k, \lambda\right) = J_r\left(\theta_{k+1}\right) - J_r\left(\theta_k\right) + \lambda\left(J_c\left(\theta_{k+1}\right) - J_c\left(\theta_k\right)\right). \tag{29}$$

With Assumption C.1, the first-order Taylor expansions of $J_r$ and $J_c$ can be written as:

$$
\begin{aligned}
J_r\left(\theta_{k+1}\right) &\leq J_r\left(\theta_k\right) + g_r\left(\theta_k\right)^T\left(\theta_{k+1} - \theta_k\right) + \frac{L_r}{2}\left\|\theta_{k+1} - \theta_k\right\|^2, \\
J_c\left(\theta_{k+1}\right) &\leq J_c\left(\theta_k\right) + g_c\left(\theta_k\right)^T\left(\theta_{k+1} - \theta_k\right) + \frac{L_c}{2}\left\|\theta_{k+1} - \theta_k\right\|^2.
\end{aligned}
\tag{30}
$$

Substituting these expansions into Eq. (29) gives:

$$
\begin{aligned}
\mathcal{L}\left(\theta_{k+1}, \lambda\right) - \mathcal{L}\left(\theta_k, \lambda\right) &\leq g_r\left(\theta_k\right)^T\left(\theta_{k+1} - \theta_k\right) + \frac{L_r}{2}\left\|\theta_{k+1} - \theta_k\right\|^2 \\
&\quad + \lambda\left(g_c\left(\theta_k\right)^T\left(\theta_{k+1} - \theta_k\right) + \frac{L_c}{2}\left\|\theta_{k+1} - \theta_k\right\|^2\right).
\end{aligned}
\tag{31}
$$

Reorganizing, we have:

$$\mathcal{L}\left(\theta_{k+1}, \lambda\right) - \mathcal{L}\left(\theta_k, \lambda\right) \leq \hat{g}_k^T\left(\theta_{k+1} - \theta_k\right) + \frac{L}{2}\left\|\theta_{k+1} - \theta_k\right\|^2, \tag{32}$$

where $L = L_r + \lambda L_c$ and $\hat{g}_k = g_r\left(\theta_k\right) + \lambda g_c\left(\theta_k\right)$.

Since the policy parameters $\theta$ are updated by gradient descent:

$$\theta_{k+1} - \theta_k = -\beta_2 h_k, \tag{33}$$

substituting this into the above inequality yields:

$$\mathcal{L}\left(\theta_{k+1}, \lambda\right) - \mathcal{L}\left(\theta_k, \lambda\right) \leq -\beta_2 \hat{g}_k^T h_k + \frac{\beta_2^2 L}{2}\left\|h_k\right\|^2. \tag{34}$$

With Assumption C.3, we have:

$$\mathcal{L}\left(\theta_{k+1}, \lambda\right) - \mathcal{L}\left(\theta_k, \lambda\right) \leq -\beta_2 \hat{g}_k^T h_k + \frac{\beta_2}{2}\left\|h_k\right\|^2. \tag{35}$$

According to Proposition D.4, we can further simplify:

$$
\begin{aligned}
\mathcal{L}\left(\theta_{k+1}, \lambda\right) - \mathcal{L}\left(\theta_k, \lambda\right) &\leq -\beta_2\left(\frac{1}{2}(1 - \rho_k^2)\|\hat{g}_k\|^2 + \frac{1}{2}\|h_k\|^2\right) + \frac{\beta_2}{2}\left\|h_k\right\|^2 \\
&\leq -\frac{1}{2}\beta_2(1 - \rho_k^2)\|\hat{g}_k\|^2 \\
&\leq 0.
\end{aligned}
\tag{36}
$$

This completes the proof. $\qquad\square$

## D.2 CONVERGENCE OF HARMONIC GRADIENT

*Proof.* The practical approach for solving the harmonic gradient in Eq. (15) is given by:

$$h_k = \hat{g}_k + \rho_k \frac{\|\hat{g}_k\|}{\|d(\alpha_k^*)\|} d(\alpha_k^*). \tag{37}$$

Using the triangle inequality as stated in Lemma D.1, we infer:

$$\|h_k\| \leq \|\hat{g}_k\| + \rho_k\|\hat{g}_k\| = (1 + \rho_k)\|\hat{g}_k\|. \tag{38}$$

Next, we apply telescoping sums from $k = 0$ to $\mathcal{K}$, where $\mathcal{K}$ denotes the number of training iterations. From Eq. (36), we have:

$$\mathcal{L}(\theta_{\mathcal{K}+1}, \lambda) - \mathcal{L}(\theta_0, \lambda) = \sum_{k=0}^{\mathcal{K}} (\mathcal{L}(\theta_{k+1}, \lambda) - \mathcal{L}(\theta_k, \lambda)) \leq -\frac{\beta_2}{2}(1 - \rho_k^2) \sum_{k=0}^{\mathcal{K}} \|\hat{g}_k\|^2. \tag{39}$$

Thus, we can bound the sum of squared gradients as:

$$\sum_{k=0}^{\mathcal{K}} \|\hat{g}_k\|^2 \leq \frac{2(\mathcal{L}(\theta_0) - \mathcal{L}(\theta_{\mathcal{K}+1}))}{\beta_2(1 - \rho_k^2)}. \tag{40}$$

Considering the minimum gradient magnitude during the training process, we have:

$$\begin{aligned} \min_{k \leq \mathcal{K}} \|\hat{g}_k\|^2 &\leq \frac{1}{\mathcal{K}+1} \sum_{k=0}^{\mathcal{K}} \|\hat{g}_k\|^2 \\ &\leq \frac{2(\mathcal{L}(\theta_0) - \mathcal{L}(\theta_{\mathcal{K}+1}))}{\beta_2(1 - \rho_k^2)(\mathcal{K}+1)}. \end{aligned} \tag{41}$$

Therefore, by applying Lemma D.3 and let the decay rate of $\beta_2$ be slower than linear, we conclude that $\min_{k \leq \mathcal{K}} \|\hat{g}_k\|^2 = o(1/\mathcal{K})$. Given Eq. (38), it follows that $\min_{k \leq \mathcal{K}} \|h_k\|^2 = o(1/\mathcal{K})$. Thus, as the number of training iterations $\mathcal{K}$ increases, the harmonic gradient $\bar{h}_k$ converges to 0.

This completes the proof. $\qquad\square$

### D.3 Convergence of HCRL

*Proof.* Given that the $\theta$ is updated by gradient descent as:

$$\theta_{k+1} - \theta_k = -\beta_2 h_k. \tag{42}$$

Taking the norm on both sides, we obtain:

$$\|\theta_{k+1} - \theta_k\| = \beta_2 \|h_k\|. \tag{43}$$

From Appendix D.2, we know that $\lim_{k \to \infty} \|h_k\| = 0$. Thus, for any $\epsilon > 0$, there exists an integer $N$ such that for all $m, n > N$,

$$\|\theta_m - \theta_n\| \leq \sum_{i=n}^{m-1} \|\theta_{i+1} - \theta_i\| = \sum_{i=n}^{m-1} \beta_2 \|h_i\| < \epsilon. \tag{44}$$

When the product of these two terms, $\beta_2$ and $\|h_i\|$, converge to zero sufficiently fast—e.g., at a faster rate than linear, the sequence $\{\theta_k\}$ satisfies the Cauchy sequence criterion as stated in Lemma D.2, which implies that $\{\theta_k\}$ converges.

This completes the proof. $\qquad\square$

## E  Proof of Optimality

### E.1 Stationarity

*Proof.* Appendix D.2 has shown that the harmonic gradient $\hat{g}_k$ converges to 0 as training progresses, i.e., $\lim_{k \to \infty} \|\hat{g}_k\| = 0$. Thus, we have:

$$\begin{aligned} \lim_{k \to \infty} \|\hat{g}_k\| &= \lim_{k \to \infty} \|g_r(\theta_k) + \lambda g_c(\theta_k)\| \\ &= \left\| g_r(\tilde{\theta}) + \tilde{\lambda} g_c(\tilde{\theta}) \right\| \\ &= 0. \end{aligned} \tag{45}$$

This implies that the stationarity condition holds, i.e.,

$$\nabla_\theta J_r\left(\tilde{\theta}\right) + \tilde{\lambda}\nabla_\theta J_c\left(\tilde{\theta}\right) = 0. \tag{46}$$

This completes the proof. □

### E.2 PRIMAL FEASIBILITY

*Proof.* We use proof by contradiction to demonstrate that primal feasibility holds. Suppose the converged solution does not satisfy the primal feasibility condition, i.e.,

$$J_c(\tilde{\theta}) > 0. \tag{47}$$

According to the update rule for the Lagrange multiplier in Eq. (23), we have

$$\lambda_{k+1} = \max\{0, \lambda_k + \beta_1 J_c(\theta_k)\}. \tag{48}$$

From Appendix D.2, we know that $\lim_{k\to\infty} \|h_k\| = 0$, which implies that $J_c(\theta_k)$ converges to a certain value $C > 0$.

As the number of training iterations approaches infinity, the Lagrange multiplier $\lambda_k$ will continuously increase due to the positive term $J_c(\theta_k)$. This contradicts the convergence of $\lambda_k$. Therefore, the assumption in Eq. (47) does not hold.

Thus, the primal feasibility condition must be satisfied:

$$J_c(\tilde{\theta}) \leq 0. \tag{49}$$

This completes the proof. □

### E.3 DUAL FEASIBILITY

*Proof.* As established in Lemma D.3, the update rule for the Lagrange multiplier ensures that it remains non-negative throughout the training process. Consequently, we can directly assert that the dual feasibility condition holds:

$$\tilde{\lambda} \geq 0. \tag{50}$$

This completes the proof. □

### E.4 COMPLEMENTARY SLACKNESS

*Proof.* We prove the complementary slackness condition by considering three cases:

1. Case 1: $\tilde{\lambda} = 0$

In this case, the complementary slackness condition naturally holds because

$$\tilde{\lambda}J_c(\tilde{\theta}) = 0 \cdot J_c(\tilde{\theta}) = 0. \tag{51}$$

2. Case 2: $\tilde{\lambda} > 0$ and $J_c(\tilde{\theta}) = 0$

Since we have established the dual feasibility condition (Appendix E.3), $\tilde{\lambda} \geq 0$. When $J_c(\tilde{\theta}) = 0$, the complementary slackness condition holds as

$$\tilde{\lambda}J_c(\tilde{\theta}) = \tilde{\lambda} \cdot 0 = 0. \tag{52}$$

3. Case 3: $\tilde{\lambda} > 0$ and $J_c(\tilde{\theta}) < 0$

Given that $J_c(\tilde{\theta}) \leq 0$ (proven in Appendix E.2), if $J_c(\tilde{\theta}) < 0$, the multiplier $\lambda_k$ will decrease according to the update rule in Eq. (23), eventually converging to 0. This scenario merges into the first case, confirming that the complementary slackness condition holds.

Since all possible cases satisfy the complementary slackness condition, we conclude that this condition holds. This completes the proof. □

# F  ENVIRONMENT

## F.1  PLANAR CONSTRAINED OPTIMIZATION

**Convex Problem.** The formulation is

$$\min_{x,y} (x - 2)^2 + (y - 3)^2$$
$$\text{s.t. } x^2 + y - 4 \leq 0.$$

Its contour plot shown in Figure 8(a) represents the level sets of the objective function, centered at $(2, 3)$, the unconstrained minimum. Darker colors indicate lower objective values, while lighter colors represent higher values. The feasible region is rendered in red. The optimal feasible point, $[1.165, 2.642]$, is where the darkest contour meets the constraint boundary, illustrating the balance between minimizing objective and satisfying constraint.

**Non-Convex Problem 1.** The formulation is

$$\min_{x,y} (x - 2)^4 - 3 (x - 2)^2 + (y - 1)^4 - (y - 1)^2$$
$$\text{s.t. } x^2 - \cos y - 1 \leq 0.$$

Its contour plot shown in Figure 8(b) represents the level sets of the objective function, centered at $(2, 1)$, the unconstrained minimum. The optimal feasible point is $[0.764, 2.0]$.

**Non-Convex Problem 2.** The formulation is

$$\min_{x,y} (x - 2)^4 - 3(x - 2)^2 + (y - 1)^4 - 2(y - 1)^2 + 2\sin(3x)\cos(3y)$$
$$\text{s.t. } x^2 + 0.5\sin(2y) - 1.2\cos(1.5x) - 1 = 0.$$

Its contour plot shown in Figure 8(c) depicts level sets of the non-convex objective together with the nonlinear equality constraint. Numerically, the optimal feasible point is $[1.128, \ 1.755]$.

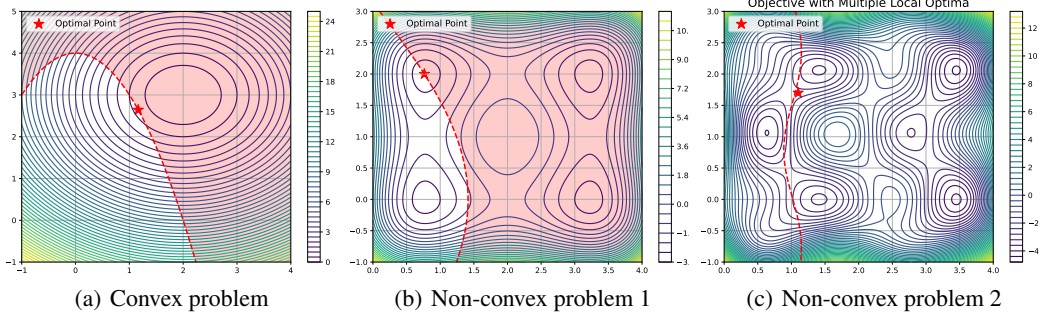

| (a) Convex problem | (b) Non-convex problem 1 | (c) Non-convex problem 2 |

Figure 8: Contour map of three planar constrained optimization problems.

## F.2 SAFETY GYMNASIUM

### F.2.1 NAVIGATION

Navigation tasks are an important class of tasks that apply RL to reality, requiring an agent to continuously change its position and interact with objects in the environment in order to accomplish a specified goal, which is usually associated with a specific position or movement pattern. In Safe RL, the focus is on the behavioral paradigm of whether an intelligent body, as a free-moving individual, can accomplish tasks in the environment without dangerous collisions or contact. In this task, the robot (in red) must reach a goal (in green) while avoiding hazards (in blue).

We consider 4 types of robots:

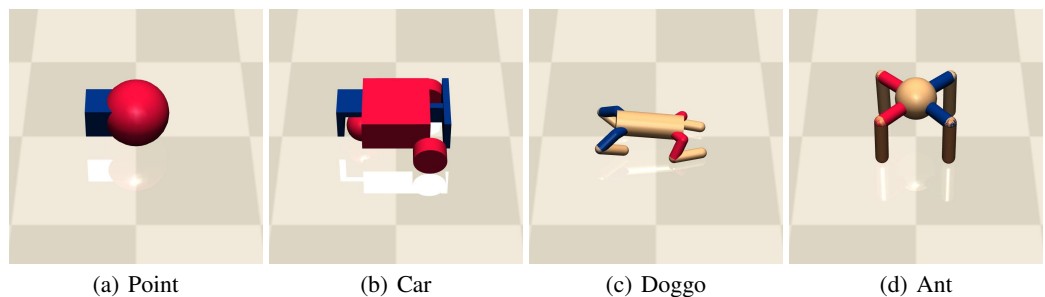

| (a) Point | (b) Car | (c) Doggo | (d) Ant |

Figure 9: Snapshots of considered 4 types of agents in goal tasks.

**PointGoal:** A simple 2D robot with two actuators, one for rotation and one for forward/backward motion. A small square in front indicates orientation and assists in pushing objects. Observation space $o \in \mathbb{R}^{12}$, action space $a \in \mathbb{R}^2$.

**CarGoal:** A slightly more complex 3D robot with two independently driven parallel wheels and a free-rolling rear wheel. Steering and forward/backward motion require coordinating the two drives. Observation space $o \in \mathbb{R}^{24}$, action space $a \in \mathbb{R}^2$.

**DoggoGoal:** A quadrupedal robot with bilateral symmetry. Each leg has two hip controls (azimuth and elevation) and one knee control (angle). Random actions allow basic travel without falling. Observation space $o \in \mathbb{R}^{12}$, action space $a \in \mathbb{R}^{12}$.

**AntGoal:** A quadrupedal robot with a torso and four legs, each with two hinged segments connected to the torso. Coordination of all eight joints is required to move in a target direction. Observation space $o \in \mathbb{R}^{40}$, action space $a \in \mathbb{R}^8$.

### F.2.2 LOCOMOTION

Velocity tasks are also an important class of tasks that apply RL to reality, requiring an agent to move as quickly as possible while adhering to velocity constraint. The Safe velocity tasks introduce velocity constraints for agents based on the Gymnasium's MuJoCo-v4 series.

**SwimmerVelocity:** The swimmer is in a 2D pool and starts at a fixed position with small uniform deviations. The goal is to move rightward as fast as possible by applying torque to the rotors, utilizing fluid friction. Observation space $o \in \mathbb{R}^8$, action space $a \in \mathbb{R}^2$.

**HalfcheetahVelocity:** A 2D robot with 9 body parts and 8 joints (including two paws). Torque is applied to six movable joints to maximize forward (right) velocity. Observation space $o \in \mathbb{R}^{17}$, action space $a \in \mathbb{R}^6$.

**HopperVelocity:** A 2D one-legged robot with torso, thigh, leg, and foot. Torque is applied to three hinges to hop forward. Observation space $o \in \mathbb{R}^{11}$, action space $a \in \mathbb{R}^3$.

**HumanoidVelocity:** A 3D bipedal robot with torso, arms, legs, and tendons. Each leg has thigh, shin, foot; each arm has upper arm, forearm. The goal is forward walking without falling. Observation space $o \in \mathbb{R}^{348}$, action space $a \in \mathbb{R}^{17}$.

**AntVelocity:** A 3D quadruped with torso and four legs, each leg with two body parts. Torque is applied to eight hinges to move forward. Observation space $o \in \mathbb{R}^{105}$, action space $a \in \mathbb{R}^8$.

**Walker2dVelocity:** A 2D bipedal robot with seven main body parts: torso, thighs, legs, and feet. Torque is applied to six hinges to walk forward. Observation space $o \in \mathbb{R}^{17}$, action space $a \in \mathbb{R}^6$.

## G  REPRODUCIBILITY STATEMENT

Our experiments are implemented on the GOPS (Wang et al., 2023) software. The hardware includes a 12th Gen Intel(R) Core(TM) i9-12900K CPU and an NVIDIA RTX 3090ti GPU. The hyper-parameter settings are listed in Table 8.

Table 8: Hyper-parameter settings.

| | Hyper-parameters | Value |
|---|---|---|
| Base | Critic learning rate $\beta_3$ | $3 \times 10^{-4} \rightarrow 1 \times 10^{-6}$ |
| | Actor learning rate $\beta_2$ | $1 \times 10^{-4} \rightarrow 1 \times 10^{-6}$ |
| | Multiplier learning rate $\beta_1$ | $1 \times 10^{-5} \rightarrow 1 \times 10^{-6}$ |
| | Replay batch size | 256 |
| | Optimizer | Adam |
| | Discount factor | 0.99 |
| | Networks | 3-layer GeLU-activated MLPs with 256 units |
| Off-policy | Number of env steps | $10^6$ (1M) |
| | Learning rate of $\alpha$ | $3 \times 10^{-4}$ |
| | Expected entropy | $-\dim(\mathcal{A})$ |
| | Policy update frequency | 2 |
| | Target update rate $\tau$ | 0.005 |
| On-policy | Number of env steps | $10^7$ (10M) |
| | Clip ratio | 0.2 |
| | GAE parameter | 0.95 |
| | Epochs per update | 10 |
| | Minibatch size | 64 |
| Lagrangian | Initial value $\lambda_0$ | 0.1 |
| | Maximum value | 20 |
| | Activation function | SoftPlus |
| | PID coefficients | $K_p = 0.1, K_I = 0.005, K_D = 0.001$ |
| HCRL | Harmonic constant $\rho_0$ | 0.9 |
| | Max iterations to solve $h$ | 20 |

## H  LLM USAGE DISCLOSURE

We used ChatGPT to polish grammar and improve text clarity. We reviewed all AI-generated suggestions and are fully responsible for the final content of this paper.

