# OpenReview forum: "Harmonic Constrained Reinforcement Learning"
_ICLR.cc/2026/Conference — ICLR 2026 Conference Withdrawn Submission_

### Official Review · Reviewer_HURv · 2025-10-15

**Soundness:** 3
**Presentation:** 3
**Contribution:** 2
**Rating:** 4
**Confidence:** 4

**Summary:**

This paper proposes HCRL that uses harmonic gradient to update policy. Instead of using lagrangian gradient, the proposed method selects a harmonic policy gradient to reduce the conflicts with reward / cost gradient within a trust region around the lagrangian gradient, which can be converted to a 1D maximization problem. The authors provide both numerical experiments and simulation results on safety gymnasium.

**Strengths:**

- Overall, the paper is clearly written and the motivation is well explained.
- The numerical experiment that compares HLag with other baselines is clear.
- The experiment on safety gymnasium is comprehensive.

**Weaknesses:**

- In this paper, the authors first set a zero-cost training objective (line 122 and 135), i.e., $J_c\leq 0$. However, almost all the learned policies in experiment has a non-zero final cost (in table 2, 6, 7), unlike other papers [1][2][3] that set a positive threshold and the final cost performance is smaller than or very close to the threshold. In other words, all those policies **do not satisify the constraint** and are not feasible. In this cases, I think the comparisons are not very meaningful.
- Even when most policies are not feasible, the HLag does not show consistent advantages on different tasks in terms of either reward maximization or cost minimization especially on on-policy method. In table 7, the PPO-HLag only achieves the lowest cost on 1 task among 10 tasks.
- Some statements are not clear. In line 182-184, "there is a conflict between ..., but h neither conflicts with gr not gc". This is true in the fig but it does not hold when $\rho$ is small or the angle between $\nabla J_r$ and $\nabla J_c$ is larger. It's not rigorous to say "h neither conflicts with gr not gc".

**Questions:**

- While HLag shows faster convergence than Lag or PIDLag in numerical experiment, the computation of harmonic gradient requires additional overhead. How does HLag compare to previous baselines in terms of computation time? E.g., within the same computation time, does Hlag still show advantage over baselines?
- The harmonic gradient is mainly used to reduce the conflicts of two objectives, which is common in multi-objective learning. How does it compare to other conflict-reduction baselines (e.g., [4][5])?

[1] Responsive Safety in Reinforcement Learning by PID Lagrangian Methods

[2] Constrained Variational Policy Optimization for Safe Reinforcement Learning

[3] OmniSafe: An Infrastructure for Accelerating Safe Reinforcement Learning Research

[4] Gradient surgery for multi-task learning

[5] Conflict-Averse Gradient Descent for Multi-task Learning

---

### Official Review · Reviewer_qix4 · 2025-10-30

**Soundness:** 2
**Presentation:** 3
**Contribution:** 1
**Rating:** 2
**Confidence:** 4

**Summary:**

The paper considers the setup of reinforcement learning with safety constraints expressed as expected cost functions over trajectories. The paper studies Lagrangian methods for training policies and proposes a new way to update the policy parameters, deviating slightly from the true gradient of the Lagrangian. The paper demonstrates empirically the advantages of this approach as compared to other methods, and also shows the convergence of the method in reinforcement learning.

**Strengths:**

The empirical results are interesting and they show advantages (fewer steps) compared to alternative methods. The evaluation includes both convex and non-convex settings to provide a sanity check that the approach works. Theoretical convergence of policy gradient with the proposed modificiation is established.

**Weaknesses:**

- The paper does not clearly justify why the proposed method for changing the update direction, as compared to the usual Lagrangian, is better. I would have expected a theoretical analysis, for example in a convex setting, showing in what regard this method is better. From my understanding, it does not necessarily find more optimal policies, but it takes fewer steps during training.
- The computational complexity is understated. In Figure 6, if I read correctly, the burden is 75% more in (a) and 35% more in (b), unlike the claimed 29-32% by the authors. It is important to note that it is not just the number of steps that matters, but also the computation effort, in judging whether an algorithm is more efficient.
- It seems the setup is established with only a single objective and a single constraint function, and I missed it whether it can accept more constraints, as is more practical.

**Questions:**

- Based on my own experience with Lagrangian methods, the tricky part is how to adapt the dual variables. On the other hand, once you have the dual variables fixed, optimizing the Lagrangian over the primal variables (policy parameters here) 'magically' makes you balance the objective and the constraints. The authors focus on the latter, and I am not sure if that is the main challenge here.
- What happens when there are multiple constraints instead of one?
- It seems that the method is designed for the case when the angle between the two gradients is larger than 90 degrees. Am I right? Is the update step unchanged when the angle is less than 90?
- Please compare with other duality works in the line of work of
Constrained reinforcement learning has zero duality gap
S Paternain, L Chamon, M Calvo-Fullana, A Ribeiro
- Lyapunov-based methods are mentioned, but they target the stricter safety problem instead of the expected cost problem, please clarify.
- Problem (1) can be written as a convex problem if the inside max over the two cases is converted to a minimization using the epigraph trick in cited Boyd's book. I hope I am not wrong. In the appendix the authors also state that this is a convex problem if I understand correctly. But why is it better to convert it to a nonconvex problem in (3)?

---

### Official Review · Reviewer_R9X6 · 2025-11-01

**Soundness:** 2
**Presentation:** 3
**Contribution:** 1
**Rating:** 4
**Confidence:** 3

**Summary:**

The paper proposes a harmonic gradient (HG) update rule for safe rl. At each policy update step, the method formulates a trust-region minimax problem: it searches for an update direction that (i) stays close to the nominal Lagrangian gradient (reward gradient plus a cost-weighted term via the Lagrange multiplier), and (ii) reduces the “worst-case conflict” between the reward objective and the safety objective. Intuitively, the algorithm tries to find a policy gradient direction that does not hurt either reward or safety too much, instead of blindly following whichever objective is currently dominant. The authors integrate the harmonic gradient update into existing CRL backbones and evaluate on ten Safety Gymnasium tasks.

**Strengths:**

The proposed harmonic-gradient update plugs into standard lagrangian baselines (e.g., PPO-Lag, DSAC-Lag) without altering their infrastructure, making adoption straightforward in existing codebases.

The paper reports results on Safety Gymnasium tasks and shows the method can achieve comparable or higher return while maintaining or reducing constraint violations.

The authors derive the  constraint optimization as a 1D problem over a mixing coefficient, which makes it inexpensive to compute in practice.

**Weaknesses:**

It remains unclear what specific deficiency in existing Lagrangian CRL the paper targets beyond “gradient conflict.” While the method reframes the update, the convergence result appears essentially unchanged relative to standard Lagrangian updates, which makes the incremental novelty hard to gauge. Please clarify the concrete failure mode addressed (with examples) and state what new theoretical or practical guarantee is gained.

In Figure 7, the curves do not include widely used safe-RL baselines such as PPO-Lag and CPO. Because these are standard points of reference, omitting them weakens the evidence. Adding PPO-Lag and CPO (with tuned hyperparameters) would make the comparison more convincing.

Even within Figure 7, the method does not clearly outperform the reported baselines across tasks. Please highlight where HCRL wins and where it does not.

The paper lacks video demonstrations in Safety Gymnasium, making it hard to assess how converged behaviors differ across methods. Providing links in the main paper (and an anonymized repo) would greatly clarify the qualitative differences between algorithms.

The provided code is not directly runnable, and the reported gains appear marginal; without a reproducible pipeline and clearer effect sizes, it is hard to assess the paper’s significance to the community.

**Questions:**

Training curves summarizing return and constraint violation over time are essential for assessing stability/robustness. Promoting representative curves to the main paper (not just appendix) would strengthen the empirical case.

PID-Lag introduces three hyperparameters (P/I/D) that can be difficult to tune consistently across tasks. The paper should also compare against PPO-Lag and CPO, which are standard references in safe RL. Did you tune PID-Lag per task? Please report these details for fair comparison.

How did you implement RCPO? Please include implementation details sufficient for replication.

Table 2 lacks per-task detail and feasibility checks. As written, it’s hard to tell on which tasks the method satisfies constraints. Please provide a per-task result in the main paper reporting (i) average return, (ii) average cost/violation rate, (iv) whether the safety budget is met (yes/no), along with variability across seeds.

Where are the cost-limit = 0 entries referenced in Table 2?

---

### Official Review · Reviewer_GXUQ · 2025-11-01

**Soundness:** 3
**Presentation:** 3
**Contribution:** 3
**Rating:** 8
**Confidence:** 3

**Summary:**

The authors propose a new, safe RL method called HCRL. This method calculates the gradients for two objectives: the return (which it aims to maximise) and the costs (which it aims to minimise). It then solves a trust region problem using an efficient transformation to find a harmonic gradient $h$, which improves both objectives while staying close to the original sum of both gradients. It demonstrates superior performance to state-of-the-art baselines on several benchmark tasks.

**Strengths:**

* Conceptually simple approach with strong mathematical foundations

* Strong empirical evaluation with lots of baselines and good results.

**Weaknesses:**

* Some related work is missing. HCRL tries to avoid conflicting gradients. There are related approaches, e.g. in multi-task RL, where we also encounter conflicting gradients for different tasks. One example: "Gradient Surgery for Multi-Task Learning", Yu et al., 2020. Presenting and comparing such works is important in order to assess the novelty of the approach.

* While the field of CRL is motivated in the introduction, HCRL is not. It would improve the paper to include some additional motivation for HCRL at the beginning.

**Questions:**

Minor mistakes:

* L77: comma missing, (2) [...] employ HG to augment Lagrangian-base CRL algorithms, off-policy DSAC
* Figure 1: The safety arrow points to the right. We would therefore expect policies on the right to be safer. Why are the worthless and risky policies then on the right? Is it flipped?
* Furthermore, I would rename "worthless" to something more descriptive, such as "low-performing" or "ineffective" policies.

Questions:

* What happens if the gradients of expected return and expected cost are almost contrary to each other, i.e. $g_j \approx - g_c$? In these cases, we cannot expect to find a harmonic update, can we?
* Figure 2: The aim is to minimize the cost, so we need to follow the negative gradient $\lambda_\theta \nabla J_c$ as $\lambda > 0$?
* How were the hyperparameters of each baseline tuned? Were they used as specified in the respective papers, or were they tuned manually per environment? Tuning can have a significant impact on performance, and a description of it often falls short. I am aware of the table of hyperparameters in the appendix.

---

### Note · Authors · 2025-11-27

I have read and agree with the venue's withdrawal policy on behalf of myself and my co-authors.